# Modeling and Optimization of Date Palm Fiber Reinforced Concrete Modified with Powdered Activated Carbon under Elevated Temperature

Musa Adamu [1], Yasser E. Ibrahim [1], Oussama Elalaoui [2], Hani Alanazi [2,*] and Nageh M. Ali [2,3]

1 Engineering Management Department, College of Engineering, Prince Sultan University, Riyadh 11586, Saudi Arabia
2 Department of Civil and Environmental Engineering, College of Engineering, Majmaah University, Al-Majmaah 11952, Saudi Arabia
3 Department of Civil Engineering, College of Engineering, Assiut University, Assiut 71511, Egypt
* Correspondence: hm.alanazi@mu.edu.sa

**Abstract:** Date palm fiber (DPF) is one of the abundant solid waste materials in the agriculture sector in Saudi Arabia, and it is gaining great attraction due to its advantages compared to synthetic and other natural fibers. For proper utilization of DPF in cementitious composites, its performance under high temperatures needs to be understood. This is because DPF is a cellulose-based agricultural fiber material and is expected to degrade when subjected to high temperatures. This will cause a significant loss in strength and structural integrity of the composites. The use of Pozzolanic materials has been reported to reduce the loss in mechanical properties of cementitious composites under high temperatures. With powdered activated carbon (PAC) being a low-cost material compared to other Pozzolanic materials, this study utilized PAC as an additive to the DPF-reinforced concrete to mitigate its loss in mechanical strength when exposed to elevated temperature. The experiment was designed using response surface methodology (RSM), which was used to construct mathematical models for estimating the strengths of the concrete exposed to high temperatures. The DPF was added at proportions of 1%, 2%, and 3% by weight of cement. Similarly, the PAC was added at 1%, 2%, and 3% by weight of cement to the concrete. The concrete was subjected to elevated temperatures of 300 °C, 600 °C, and 900 °C for a 2 h exposure period. The degradation of the concrete in terms of mass loss and the compressive strength of the concrete after heating were measured. DPF in the concrete led to an escalation in weight loss and reduction in strength, which was more pronounced at a temperature of 600 °C and above. The addition of PAC resulted in an enhancement in the strengths of the concrete containing up to 2% DPF at 300 °C, while at 600 °C the improvement was minimal. The models developed for predicting the mass loss and strengths of the DPF-reinforced concrete under high temperatures were statistically significant with a high correlation degree. Based on the optimization results, DPF-reinforced concrete produced with 1% DPF, and 2.27% PAC as additives and subjected to a temperature of 300 °C for 2 h yielded the lowest mass loss of 2.05%, highest residual compressive strength and relative strength of 45.85 MPa and 106.7% respectively.

**Keywords:** date palm fiber; powdered activated carbon; elevated temperature; response surface methodology; mass loss; compressive strength

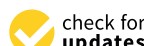



## 1. Introduction

Worldwide, enormous amounts of date palm fibers (DPF) are generated and discarded every year. The utilization of this natural fiber in the construction industry as a raw material would help to value more natural sustainable resources. The construction industry is currently dealing with significant energy and environmental issues [1]. To slow environmental deterioration and prevent an early depletion of energy resources, researchers should be encouraged to adopt the utilization of natural sustainable materials with low

environmental impact. DPFs are environmentally friendly alternatives compared to mineral or synthetic ones. Incorporating DPFs into the concrete mixtures will contribute to increasing thermal and acoustic insulation and enhance the resistance of concrete structures to elevated temperatures.

Concrete structures are built for normal as well as exposure environments such as high temperatures, acidic environments, chlorine environments, etc. One of the main advantages of concrete over other construction materials is its high resistance to high temperatures without loss of its integrity [2]. When a concrete structure is exposed to elevated temperatures in areas such as in the nuclear industry for prestressed concrete pressure vessel or in power plants as cooling towers, the mechanical strengths and structural integrity of the concrete is at risk of deteriorating due to the effect of heat [3]. Therefore, knowledge of the strengths, toughness, brittleness, durability, and correlation between the residual strengths and elevated temperature is necessary as it will aid in identifying and assessing the damage levels caused to the concrete after exposure to high temperatures [3]. The performance of concrete under elevated temperatures is reliant on the composition, type, and content of the constituent materials used, especially the aggregate types and binders. The type and quantities of fiber added to the concrete also influence its performance when subjected to high temperatures [4,5]. Hence it is very essential to be careful when selecting the constituent materials to produce concrete that might be exposed to high temperatures to limit the damaging effect of the heat. The influence of high temperatures on concrete occurs when the temperature exceeds 110 °C which leads to dehydration and escape of chemically bonded water from the calcium-silicate hydrate (C-S-H). When the temperature reaches about 300 °C, thermal expansion of the aggregates and dehydration begin to occur leading to microcrack formation within the concrete matrix. At about 400 °C, decomposition of $Ca(OH)_2$ takes place, while at above 600 °C C-S-H decomposition begins. The continuous decomposition of the hydration products becomes more severe at about 800 °C, leading to a significant decline in the mechanical strengths and structural integrity of the concrete [6–9]. Different types of fibers have been used to enhance the tensile behavior and increase the ductility of cementitious composites. The fibers are either natural or synthetic. Natural fibers including plant-based and animal-based fibers are less expensive and more environmentally friendly. However, they do not significantly improve the performance of cement composites. Synthetic fibers including steel fibers, glass fibers, carbon fibers, and asbestos are more expensive and mostly not eco-sustainable compared to natural fibers but significantly enhance the performance of composites [10]. Other new fibers have been developed from waste materials and have been found to improve the fracture toughness and tensile strength of concrete. These fibers are generated from nylon waste, carpets, scrap tires, and plastic sacks [11]. Fibers in concrete have been reported to enhance their resistance to high temperatures propagation of thermal cracks and preventing spalling in the concrete [12,13]. The application of natural fiber to cementitious composites has contributed significantly to green construction and an eco-sustainable environment. This is because natural fibers are non-toxic, and their processing consumes lower energy compared to synthetic fibers such as steel, glass, or carbon. Additionally, their usage in cementitious composites does not cause any health hazards such as synthetic fibers such as asbestos. However, natural fibers especially agriculturally based due to their organic nature, hydrophobicity, high cellulose, and lignin contents, when used as a fiber in concrete give the concrete high vulnerability to elevated temperature or fire [14–16]. The effects elevated temperatures oh the properties of natural cellulose fiber reinforced concrete have been studied. Zhang, Tan [17] investigated the effect of jute fiber (JF) on the heat and spalling resistance of ultra-high-performance concrete (UHPC). The UHPC was prepared by adding three proportions of JF: 3, 5, and 10 kg/m³. Then, the specimens were subjected to varying temperatures of 200 °C, 400 °C, 600 °C and 800 °C for 2 h exposure. The findings showed that the JF shrunk with increase in temperature which increases the permeability of the UHPC at high temperature and consequently aids in resisting thermal spalling. With regards to compressive strength, addition of 3 kg/m³ JF led to enhancement in the strength at 200 °C and 400 °C, while

5 kg/m$^3$ and 10 kg/m$^3$ JF addition resulted in reduction in the strength of UHPC at all temperature, with the decline more pronounced at higher temperatures. Th thermal properties of JF reinforced high strength concrete were examined in Ozawa, Kim [18]. The concrete specimens were prepared using 0% and 0.075% JF by volume of the concrete. The concrete was subjected to temperatures of 100 °C to 500 °C. The results showed that the concrete reinforced with JF losing 40% of its strength at 100 °C, then recovered its full strength between 200 °C to 300 °C compared to the plain concrete. They concluded that JF can be utilized for the prevention of explosive spalling in high strength concrete without enhancing its strength. Gonzalez-Lopez, Claramunt [19] subjected fiber reinforced calcium aluminate cement-based composite containing 5% nonwoven flax fabrics fibers (NFFF) by weight of the composite to temperatures of 250 °C, 450 °C, and 950 °C. They reported a reduction in mechanical performance with an increase in temperature. They found that the NFFF maintained its reinforcement capacity when the composite was exposed to heat up to 250 °C where the composites maintained their mechanical performance. However, when the composite was exposed to 450 °C heat, there was a significant decline in strength due to the NFFF losing its integrity. Wei and Meyer [20] studied the degradation mechanism of natural fiber (sisal fiber) in the cementitious composite matrix. 2% of sisal fiber was added to the cement mortar containing 10% to 30% metakaolin as a partial replacement for cement. Serious degradation of the fiber in the cement matrix at 300 °C was observed, however at lower temperatures between 260 to 280 °C, the degradation of the fiber in the matrix was gradual. It was noted continual mass loss at high temperatures due to the thermal decomposition of lignin and cellulose of the fiber. Grubeša, Marković [21] investigated the influence of hemp fiber-reinforced concrete under elevated temperatures. 2.5% of hemp fiber was added to the concrete and subjected the concrete to temperatures of 20 °C and 400 °C. An improvement in compressive strength with the addition of hemp fiber at 20 °C was reported, while at 400 °C, there was a loss in compressive strength of about 46.1%. It was attributed the loss in strength at high temperatures to the degradation of the fiber and cement matrix.

Pozzolanic materials were used to improve the performance of fiber-reinforced concrete under high temperatures. Gencel, Nodehi [22] replaced 15% cement with silica fume in basalt fiber reinforced foam concrete (BFRPC) containing 0%, 1.5%, and 3% basalt fiber. They reported improvement in residual compressive strength with the addition of silica fumes at up to 800 °C. In comparison to the BFRFC with 0% silica fume, the concrete with 15% silica fume recorded an increase in strength up to 129% at 800 °C, whereas at lower temperatures the increment was more. They attributed this enhancement to the higher thermal resistance and high pozzolanic reactivity of the silica fume in comparison to cement. Jameel, Raza [23] partially replaced 40% cement with fly ash in polypropylene fiber reinforced concrete (PPFRC) containing 0.5% fiber. They subjected the PPFRC to high temperatures of 100, 200, 350, 450, and 650 °C. They reported enhancement in compressive strength of the PPFRC with the addition of fly ash at temperatures up to 200 °C, which they attributed to the thermal stimulation and faster glass breakdown of the fly ash compared to cement. Yonggui, Shuaipeng [24] partially replaced cement with nano silica (NS) in BFRC and study the mechanical performance of the concrete at 25, 200, 400, and 600 °C. They added the fiber at dosages of 0, 1, 2, and kg/m$^3$ and NS proportions were 0%, 3%, 6%, and 8% by volume of cement. It was reported that NS helped enhance the compressive strength of the BFRC at any temperature, and contributed to the enhancement of the ITZ performance at elevated temperatures. Mahapatra and Barai [25] also reported improvement in the strength of hybrid fiber reinforced concrete (steel and polypropylene fibers) containing fly ash as a partial substitute to cement and colloidal NS as an additive to cement at up to 600 °C and without the occurrence of any spalling.

Powdered activated carbon (PAC) is an amorphous-carbonaceous substance with characteristics such as larger surface areas, adsorptive capacity, and porous texture [26]. PAC is a micro-porous black powder usually derived mainly from charcoal or waste products from coconut or rice husks combustion. Due to the excellent characteristics

of PAC, it is typically employed to investigate functional absorption in industrial and cementitious materials [27]. However, there are only limited studies on the effect of PAC on concrete properties. It was found that adding activated carbon in concrete mixtures increased $NO_2$ absorption from the atmosphere without compromising the mechanical properties or increasing overall porosity [28,29]. PAC has large surface areas and high porosity even at the nanoscale, making them a suitable constituent material for densifying the microstructure and refining the porosity of the cement matrix [30]. Previous studies have shown that PAC when used as an additive to cementitious composites such as concrete and mortar improved the mechanical strengths and reduced the curing time required to achieve the desired strength. Additionally, due to its large surface area, PAC acts as a filler and densified the microstructure of composites [31,32].

DPF is a natural fiber derived from date palm tree mesh which is in abundance in Middle Eastern and North African Countries. DPF has some advantages over synthetic and other natural fibers which include low to zero cost, ease of processing, and a higher strength-to-cost ratio [10,33,34]. The inclusion of DPF as a natural fiber in cementitious composites was reported to improve its thermal insulation and acoustic properties [35–37]. However, the main shortcoming of adding DPF in concrete mixtures is the reduction in mechanical strength as reported by several studies. This reduction is mainly caused due to an increase in pore volume in the cement matrix because of the hydrophilic nature of DPF [10,38,39]. To reduce the negative effect of the DPF on the strengths of concrete, researchers treat the DPF mostly using an alkaline solution to increase the surface roughness in order to enhance the bonding between the cement paste and DPF. Additionally, cementitious additives such as silica fume were typically included in DPF concrete to densify its microstructure and enhance its strength [10,22].

DPF-reinforced concrete has been found to have some advantages such as lower density and higher thermal insulation. However, when DPFRC is subjected to areas of high-temperature application, there is a high tendency for the fiber to degrade at very high temperatures due to its vegetative and organic nature, and high cellulose and lignin content. This can cause a significant loss in strength and mass, and consequently, affect the structural integrity of the structure constructed using DPF-reinforced concrete. However, there is very scanty literature that studied the performance of DPF-reinforced concrete under elevated temperatures. Additionally, PAC is a cheaper material compared to other pozzolanic materials such as silica fume, and its use has been reported to enhance the mechanical performance of cementitious composites. Hence, this study utilized PAC as an additive in DPF-reinforced concrete to improve the concrete's performance under elevated temperature, for which to the best knowledge of the authors, there is no available study that used PAC to improve the performance of DPFRC concrete under high temperature. Therefore, this study utilized PAC to mitigate the undesirable effects of DPF on the strengths of concrete at high temperatures.

Response surface methodology (RSM) analysis technique was employed for designing the experimental mixes, the establishment of model equations for the prediction of the strengths of the DPFRC under high temperature, and optimization of the mix proportions to achieve the optimum combinations of the variables that will yield the best performance of the concrete under high temperature. This study utilized PAC to mitigate the undesirable effects of DPF on the strengths of concrete at high temperatures.

## 2. Materials and Methods

### 2.1. Materials

Investigations were conducted on concrete made of cement type CEM I 42.5 R confirming the specifications of ASTM C150/C150M [40]. The cement with a specific gravity of 3.15 and a bulk density of 1440 kg/m$^3$ was completely dry and free from lumps. The incorporated PAC was supplied and manufactured by BMS Factories (Gharbalah industrial company in the Kingdom of Saudi Arabia). The PAC added in terms of the mass of the binder has an iodine number of 1450 mg/g, bulk density of 0.55 g/cm$^3$, a specific surface

area of 3000 m²/g, and an average pore size of 2.14 nm. The fine and coarse aggregates used were natural river sand and crushed granites respectively. Both types of aggregates were cleaned before being added to cement paste in saturated surface dried (SSD) condition. The aggregates' gradation is depicted in Figure 1, while their physical properties are summarized in Table 1. A polycarboxylate-based superplasticizer having a density of 1060 kg/m³ was used as a water reducer.

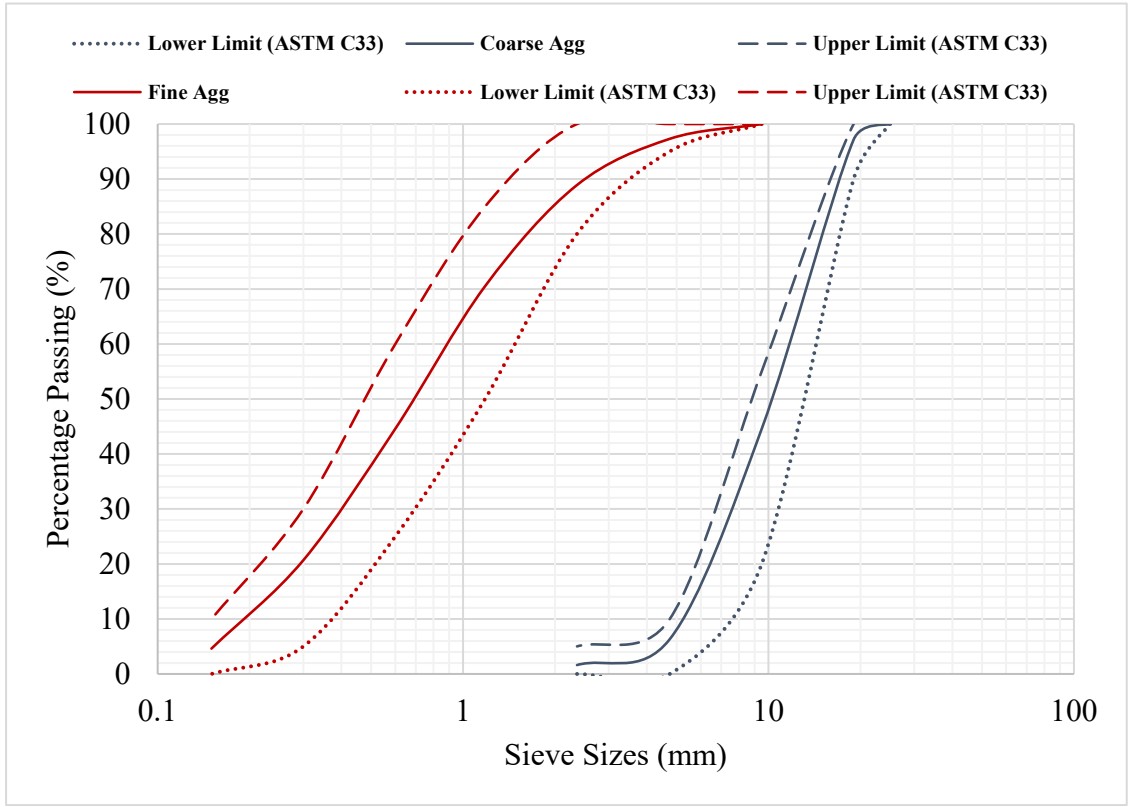

**Figure 1.** Particle size distribution of aggregates.

**Table 1.** Physical properties of aggregate.

| Property | Fine Aggregate | Coarse Aggregate |
|---|---|---|
| Specific gravity | 2.63 | 2.67 |
| Bulk density (kg/m³) | 1560 | 1460 |
| Fineness modulus | 2.3 | - |
| Water absorption (%) | 1.9 | 0.7 |
| Condition | SSD | SSD |

DPF is a natural date palm fiber collected in its raw state from a nearby date farm. DPF has a rectangular interwoven mesh of about 30–50 cm in length by 20–30 mm in width as shown in Figure 2a. The raw fiber was prepared before being used in concrete by soaking it in water for about 2 h followed by washing with clean water. After that, fibers were immersed in a 3% NaOH solution for three hours to increase fiber roughness by removing any impurities or dust from the surface. The drying process is maintained between 48 and 72 h until ensuring that fibers are completely dried. The final preparation step consists of the manual separation of fiber bundles into single fibers of about 2–3 cm in length and 0.02–0.1 cm in diameter as shown in Figure 2b. In this study, cement-based concrete incorporation of single DPF with fiber content ratios of 1%, 2%, and 3% by cement mass was investigated.

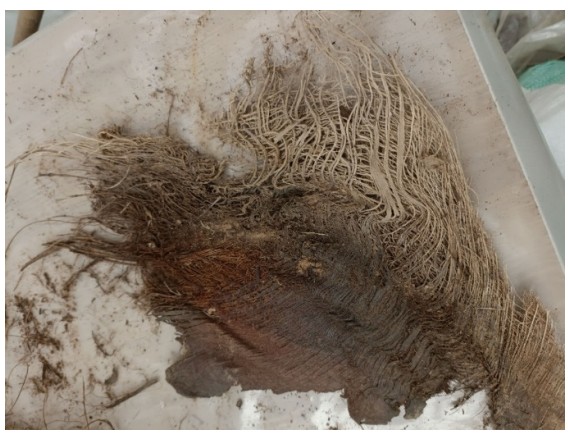
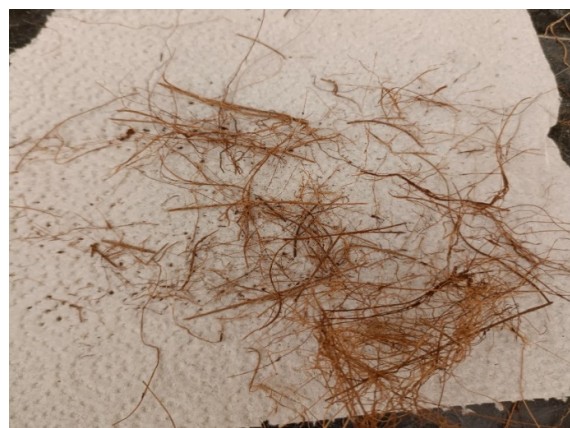

(**a**) DPF in interwoven form                    (**b**) Single DPF

**Figure 2.** Particle size distribution of aggregates.

### 2.2. Mix Proportioning by Response Surface Methodology (RSM)

RSM was employed in this study, and it is a widely used approach to designing experiments and optimizes the effect process variables via a set of statistical and mathematical tools [41]. This approach was employed by the inclusion of independent factors establishing properties of DPF-reinforced concrete after exposure to elevated temperatures. RSM relates a response to the levels of several input variables or factors that influence it.

Numerous model types are applied in RSM depending on the number of independent variables and their proportions. Models such as $2^k$ factorial design, Box–Behnken design (BBD), D-optimal design, central composite design (CCD), Taguchi's OA design, Historical data design, user-defined, miscellaneous design, and one-factor design method are available [31,32]. RSM is implemented in three steps: (a) experiment design such as Central Composite Design; (b) statistical and regression analysis to build equations that describe well the modeling of the response surface; and (c) optimization process of independent variables carried out through the model Equation. These model equations are in the form of either linear as given in Equation (1a) or higher order polynomials as shown in Equation (1b). In most cases, the linear equation type is not suitable due to the presence or existence of curvature [41].

$$\mathcal{R} = \acute{z}_0 + \acute{z}_1 V_1 + \acute{z}_2 V_2 + \dots \acute{z} V_n + \xi, \tag{1a}$$

$$\mathcal{R} = \acute{z}_0 + \sum_{x=1}^{n} \acute{z}_i V_x + \sum_{x=1}^{n} \acute{z}_{xx} V_x^2 + \sum_{x<} \sum_{y} \acute{z}_{xy} V_x V_y + \xi \tag{1b}$$

where $\mathcal{R}$ is the response of variables, $\acute{z}_0$ is the intercept for which $V_1 = V_2 = 0$, $\acute{z}_1$ is the coefficient of the 1st variable, $\acute{z}_2$ is the coefficient of the 2nd variable, $x$ is the linear implicit value for the variable, $y$ is the quadratic implicit value for the variable, $\xi$ is the error, and $n$ is a number of variables.

In this investigation, the RSM analysis was carried out using Design Expert software. The BBD model is considered one of the most appropriate methods that can be specially designed to fit a second-order model. The three studied variables are DPF content, PAC content, and three levels of temperature. The effects of three different contents of DPF and PAC (1%, 2%, and 3%) by weight of cement and exposure temperatures (300 °C, 600 °C, and 900 °C) for a period of exposure of 2 h are evaluated.

The control concrete was produced according to ACI 211.1R [42] via the method of absolute volume. The concrete mix was designed based on a target cube compressive strength of 37 MPa at 28 days using a water/cement ratio of 0.38. Based on the number of variables and their proportions, the concrete mix design was established by the RSM using the proportions of the independent variables. Nineteen (19) mixes were selected as presented in Table 2. The central mix (mix containing 2% DPF, 2% PAC, and 600 °C)

was replicated five times, and the variation of the results was used to compute the lack of fit relative to the pure error. The mixes were fabricated and allowed to cure for 28 days in controlled conditions till being exposed to elevated temperatures for two hours and then tested.

**Table 2.** Mix proportioning based on RSM.

| Run | Variables | | | Quantity (kg/m³) | | | | | | | |
|---|---|---|---|---|---|---|---|---|---|---|
| | DPF | PAC | Temperature (°C) | Cement | PAC | Fiber | Fine Aggregate | Coarse Aggregate | Water | S. P |
| Control | 0 | 0 | - | 480 | 0 | 0 | 730 | 890 | 180 | 4.8 |
| 1 | 1 | 2 | 300 | 480 | 9.6 | 4.9 | 730 | 890 | 180 | 4.9 |
| 2 | 2 | 2 | 600 | 480 | 9.6 | 9.8 | 730 | 890 | 180 | 4.9 |
| 3 | 3 | 2 | 900 | 480 | 9.6 | 14.7 | 730 | 890 | 180 | 4.9 |
| 4 | 1 | 1 | 600 | 480 | 4.8 | 4.8 | 730 | 890 | 180 | 4.9 |
| 5 | 2 | 2 | 600 | 480 | 9.6 | 9.8 | 730 | 890 | 180 | 4.9 |
| 6 | 2 | 3 | 300 | 480 | 14.4 | 9.9 | 730 | 890 | 180 | 4.9 |
| 7 | 1 | 3 | 600 | 480 | 14.4 | 4.9 | 730 | 890 | 180 | 4.9 |
| 8 | 3 | 3 | 600 | 480 | 14.4 | 14.8 | 730 | 890 | 180 | 4.9 |
| 9 | 2 | 2 | 600 | 480 | 9.6 | 9.8 | 730 | 890 | 180 | 4.9 |
| 10 | 2 | 2 | 600 | 480 | 9.6 | 9.8 | 730 | 890 | 180 | 4.9 |
| 11 | 2 | 1 | 900 | 480 | 4.8 | 9.7 | 730 | 890 | 180 | 4.9 |
| 12 | 1 | 2 | 900 | 480 | 9.6 | 4.9 | 730 | 890 | 180 | 4.9 |
| 13 | 2 | 3 | 900 | 480 | 14.4 | 9.9 | 730 | 890 | 180 | 4.9 |
| 14 | 2 | 2 | 600 | 480 | 9.6 | 9.8 | 730 | 890 | 180 | 4.9 |
| 15 | 3 | 1 | 600 | 480 | 4.8 | 14.5 | 730 | 890 | 180 | 4.9 |
| 16 | 2 | 1 | 300 | 480 | 4.8 | 9.7 | 730 | 890 | 180 | 4.9 |
| 17 | 3 | 2 | 300 | 480 | 9.6 | 14.7 | 730 | 890 | 180 | 4.9 |

*2.3. Sample Preparations and Casting*

In this study, the procedure in ASTM C192/C192M [43] which covers batching, sampling, mixing, and curing of the concrete is followed. Concrete samples were prepared in three steps. First, the cement, PAC, and fine aggregate were mixed in a rotating drum concrete mixer. After that, the DPF and coarse aggregate were added to the mixer and thoroughly mixed. Finally, the water plus superplasticizer was first mixed and then added to the mixer progressively as the mixing continued. After all, the ingredients are added to the mixer, concrete is mixed for 3 min, followed by 3 min rest, followed by 2 min of final mixing. The freshly mixed concrete was then poured into molds. Mixtures are placed in three consecutive layers and compacted separately. The surface of the fresh concrete is finished and leveled. Samples are allowed to harden for a period of 24 h before being demolded. Samples are then cured at controlled temperature and humidity until the testing age is reached.

*2.4. Test Methods*

Cubic samples of 100 mm sizes were prepared and cured for 28 days in water prior to testing. After curing, the concrete specimens were air-dried for at least 24 h before being tested. The weight of samples was taken as $W_0$ and samples were then heated using an electrical furnace for different setting temperatures (300 °C, 600 °C, 900 °C for a 2 h exposure period) as prescribed by the RSM. The electric furnace has a capacity of 1300 °C. The temperature was applied gradually at a constant increment rate of 10 °C/min until the desired temperature is reached and then maintained at a constant temperature for 2 h before the furnace was switched off. Figure 3a shows the samples inside the furnace after the maximum temperature was reached. The samples were kept in the furnace and kept closed until the samples were completely cooled. Figure 3b presents the cooled samples inside the furnace, while Figure 3c presents the samples heated at 600 °C after cooling. The

heated samples were weighed after extraction from the furnace and recorded as $W_T$. The mass loss was then calculated using Equation (2).

$$\gamma_L(\%) = \frac{W_0 - W_T}{W_0} \times 100 \tag{2}$$

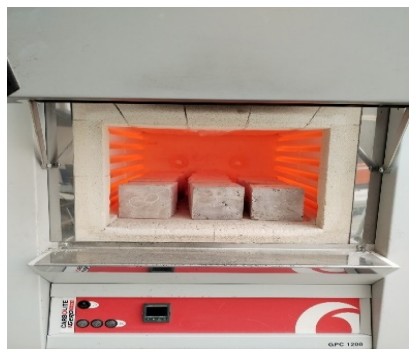
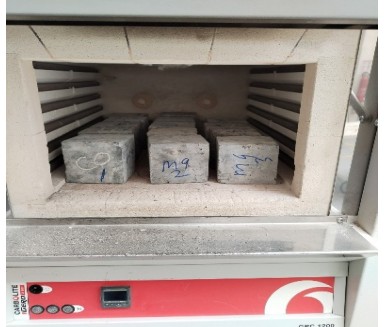

(**a**) Heated Samples      (**b**) Cooled Samples

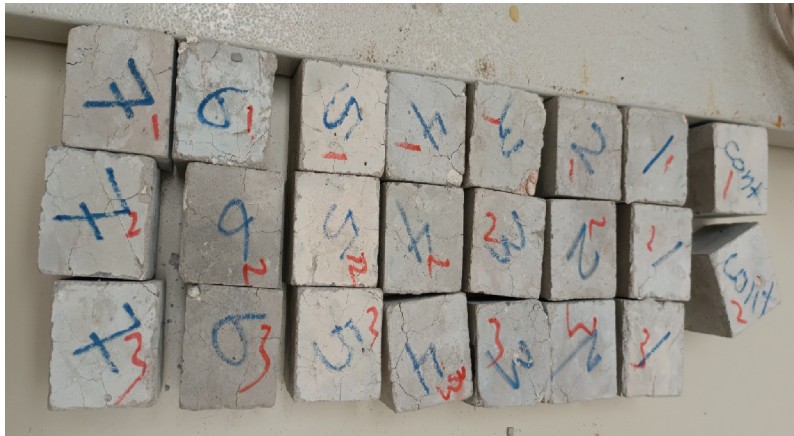

(**c**) Samples after heating at 600 °C and cooled

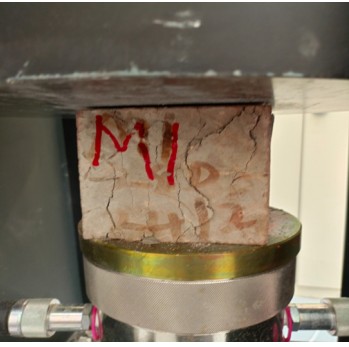
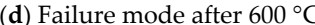
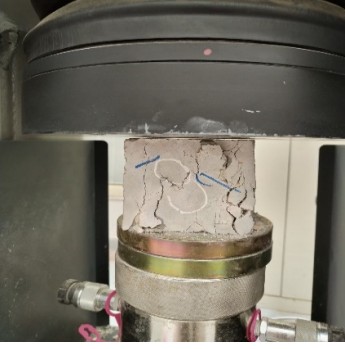

(**d**) Failure mode after 600 °C      (**e**) Failure mode after 900 °C

**Figure 3.** Samples heating and testing.

$\gamma_L$ represents the mass loss in %, $W_0$ represents the initial sample weight before heating in grams, and $W_T$ represents the final sample weight after temperature exposure in grams.

The compressive testing was performed using a 2000 kN universal testing machine according to BS EN 12390-3 [44]. The UTM was set to a pacing rate of 2.5 kN/s corresponding to a stress rate of 0.25 MPa/s. Figure 3d,e presents the failure modes of the samples after heating at 600 °C and 900 °C respectively, under compressive loads. The relative strength of each mix is calculated using Equation (3). For each of the mixes, three samples were

prepared and the average results were recorded. Figure 4 summarizes the mythology in the form of a flow chart.

$$F_R = 100 - \left[ \left( \frac{F_{c,0} - F_{c,T}}{F_{c,0}} \right) \times 100 \right] \qquad (3)$$

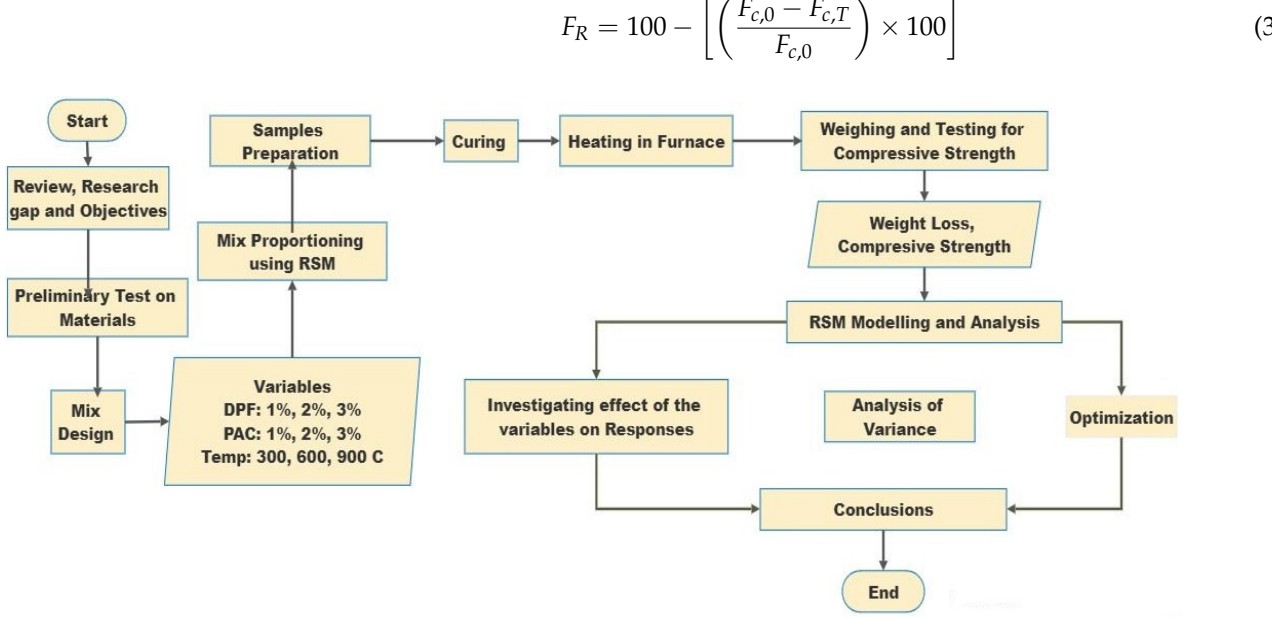

**Figure 4.** Flow chart of methodology.

In Equation (3), $F_R$ represents the relative strength in %, $F_{c,0}$ represents the 28 days compressive strength before being heated in MPa, and $F_{c,T}$ represents the compressive strength after exposure to the setting temperature, expressed in MPa.

## 3. Results and Discussions

The changes in the mass and compressive strength of the DPF-reinforced concrete containing PAC due to exposure to elevated temperatures is illustrated in Table 3. These results are used hereafter for developing mathematical models and for optimization using the RSM approach.

**Table 3.** Results of mass loss and compressive strength after heating.

| Run | Variables | | | Responses | | | |
|---|---|---|---|---|---|---|---|
| | A: DPF | B: PAC | C: Temperature (°C) | Mass Loss (%) | Residual Compressive Strength (MPa) | Normal Compressive Strength (MPa) | Relative Strength (MPa) |
| 1 | 1 | 2 | 300 | 2.41 | 47.01 | 44.32 | 106.07 |
| 2 | 2 | 2 | 600 | 6.92 | 18.15 | 42.15 | 43.06 |
| 3 | 3 | 2 | 900 | 6.22 | 18.19 | 41.88 | 43.16 |
| 4 | 1 | 1 | 600 | 4.68 | 31.24 | 54.13 | 57.71 |
| 5 | 2 | 2 | 600 | 6.74 | 23.12 | 45.22 | 51.13 |
| 6 | 2 | 3 | 300 | 3.74 | 22.65 | 31.95 | 70.89 |
| 7 | 1 | 3 | 600 | 6.65 | 30.63 | 38.09 | 80.41 |
| 8 | 3 | 3 | 600 | 5.74 | 15.35 | 24.53 | 62.58 |
| 9 | 2 | 2 | 600 | 6.19 | 20.64 | 40.05 | 51.54 |
| 10 | 2 | 2 | 600 | 7.01 | 24.65 | 46.1 | 53.47 |
| 11 | 2 | 1 | 900 | 10.24 | 9.6 | 54.13 | 17.73 |
| 12 | 1 | 2 | 900 | 9.56 | 11.82 | 44.32 | 26.67 |
| 13 | 2 | 3 | 900 | 10.57 | 7.42 | 31.95 | 23.22 |
| 14 | 2 | 2 | 600 | 6.88 | 19.45 | 39.63 | 49.08 |
| 15 | 3 | 1 | 600 | 3.95 | 31.81 | 47.48 | 67 |

**Table 3.** *Cont.*

| Run | Variables | | | Responses | | | |
|---|---|---|---|---|---|---|---|
| | **A: DPF** | **B: PAC** | **C: Temperature (°C)** | **Mass Loss (%)** | **Residual Compressive Strength (MPa)** | **Normal Compressive Strength (MPa)** | **Relative Strength (MPa)** |
| 16 | 2 | 1 | 300 | 2.62 | 30.4 | 51.39 | 59.16 |
| 17 | 3 | 2 | 300 | 1.98 | 25.56 | 42.96 | 59.5 |

*3.1. Mass Loss*

3.1.1. Analysis of Variance for Mass Loss

Table 4 summarized the analysis of variance (ANOVA) results for the statistical models developed for predicting the mass loss of the DPF-reinforced concrete exposed to different temperature exposure. The confidence interval i.e., probability (*p*-value) less than 0.05 ($p < 0.05$) was used to test the significance of the model and all its terms. The mass loss model is statistically significant as it has a lower *p*-value which is quite less than 0.05. Therefore, the null hypothesis must be rejected, and a relationship existed between the mass loss and the variables (DPF, PAC, and temperature). A quadratic model fitted the data the best was chosen. The letters D, A, and T were used to denote DPF, PAC, and temperature respectively in all the models. The significance of each of the terms in the model was also assessed using $p < 0.05$. The mass loss model significant terms are D, A, T, and $D^2$ all of which have *p* values less than 0.05. While the terms D*A, D*T, A*T, $A^2$, and $T^2$ were not significant in the model as their *p* values are more than 0.05. A further significant check was conducted on the lack of fit of the model. A significant lack of fit implies a non-fitted model with a problem with the experimental data and or the model. From Table 4, the lack of fit is significant as its *p*-value is less than 0.05. This explained there is a problem with the model or data. Hence, the model depicted in Equation (4a) cannot be used to predict the mass loss for the concrete. To address this problem, either model reduction by eliminating the non-significant terms in the model or model transformation needs to be completed. Hence, model reduction through the backward elimination method was completed. From Table 4 it can be seen that after model reduction, the lack of fit became non-significant. Therefore, the model presented as Equation (4b) is the most appropriate for estimating the mass loss of the DPF-reinforced concrete under elevated temperature.

$$W_L = -9.85 + 7.355^*D + 0.622^*A + 0.0181^*T - 0.045^*D^*A - 0.0024^*D^*T - 0.0007^*A^*T - 1.622^*D^2 + 0.129^*A^2 - 0.0000009^*T^2 \tag{4a}$$

$$W_L = -9.03 + 7.255^*D + 0.651^*A + 0.0156^*T - 0.00243^*D^*T - 1.62^*D^2 \tag{4b}$$

Table 5 gives the summary for further ANOVA for the mass loss model. The degree of determination ($R^2$) values were used to further explain and validate the models. A well-fitted model will have a very high $R^2$ value close to unity (one), while a poorly fitted model will have very low $R^2$ values near zero. The mass loss model has an excellent $R^2$ number of 0.974, implying that only around 2.6% of the investigational data does not completely fit the model. After model reduction, the $R^2$ value became 0.971 which is also very high. Moreover, the disparity between the predicted and adjusted $R^2$ was used for further model validation statistically. For a good model, the disparity must be less than 0.2. If greater than 0.2, model reduction or transformation needs to be carried out to fix the error with the model and or data. From Table 5, before model reduction, the disparity between the predicted and adjusted $R^2$ for the mass loss was greater than 0.2. This further justified problem with the model or experimental data. However, after model reduction, the disparity between the two $R^2$ values became less than 0.2. Further ANOVA checks with regard to the standard deviation relative to mean value, coefficient of variation, and adequate precision was carried out. The less standard deviation compared to the mean explained the lesser variability of the experimental data relative to the model which was

better after model reduction. The model for the mass loss is effective for the navigation of the design space as its adequate precision is higher than 4 [45].

**Table 4.** ANOVA summary for mass loss model.

| | Before Reduction | | | | After Reduction | | |
|---|---|---|---|---|---|---|---|
| Source | F Value | *p*-Value Prob > F | Significance | Source | F Value | *p*-Value Prob > F | Significance |
| Model | 28.56 | 0.0001 | Yes | Model | 73.84 | <0.0001 | Yes |
| D-DPF | 9.04 | 0.0197 | Yes | A-DPF | 13.02 | 0.0041 | Yes |
| A-PAC | 8.39 | 0.0231 | Yes | B-PAC | 12.08 | 0.0052 | Yes |
| T-Temperature | 206.33 | <0.0001 | Yes | C-Temperature | 297.07 | <0.0001 | Yes |
| D*A | 0.020 | 0.8915 | No | — | — | — | — |
| D*T | 5.23 | 0.0560 | No | AC | 7.54 | 0.0191 | Yes |
| A*T | 0.39 | 0.5542 | No | — | — | — | — |
| $D^2$ | 27.37 | 0.0012 | Yes | $A^2$ | 39.51 | <0.0001 | Yes |
| $A^2$ | 0.17 | 0.6909 | No | — | — | — | — |
| $T^2$ | 0.073 | 0.7942 | No | — | — | — | — |
| Lack of Fit | 7.51 | 0.0404 | Yes | Lack of Fit | 3.56 | 0.1184 | No |

**Table 5.** Mass loss model validations.

| Term | Before Reduction | After Reduction |
|---|---|---|
| $R^2$ | 0.974 | 0.971 |
| Adjusted $R^2$ | 0.939 | 0.958 |
| Predicted $R^2$ | 0.634 | 0.893 |
| Adequate Precision | 17.89 | 27.89 |
| Standard Deviation | 0.64 | 0.53 |
| Mean | 6.01 | 6.01 |
| Coefficient of variation (%) | 10.59 | 8.83 |

3.1.2. Diagnostic Plots for Weight Loss Model

The predicted (from the model) versus actual (experimental data) plots for the mass loss model are presented in Figure 5a. A very high correlation exists between the predicted and actual mass loss as verified graphically, where the data points fall and correlated within the straight trend line drawn. This validates the predictive ability of the model. Figure 5b depicts the normal probability versus studentized residuals for the mass loss model. Initially, during the design of the experiment, model establishment, and examination, it was presumed that the experimental data and model followed the normal probability distribution. Therefore, there is a need to check and confirm. From Figure 5b, the data points were well correlated within the straight line, therefore the mass loss model is said to be normally distributed.

To measure how each variable at a certain reference point affects the responses, the perturbation plot was plotted for the mass loss model as shown in Figure 6. The perturbation plot shows how one variable moves beyond the reference point while the other independent variables are kept constant at that location [45]. In the perturbation plot, A stands for DPF, B is PAC, and C represents temperature. Lines B and C tend to be flatter (straight), while line A has more curvature. Therefore, the DPF has more sensitivity to mass loss under high temperatures compared to PAC and temperature change. While based on perturbation, the PAC and temperature have less sensitivity to the mass loss of the DPF-reinforced concrete. The steep positive slope of C i.e., temperature means that the temperature is significantly contributing to the increase in mass loss.

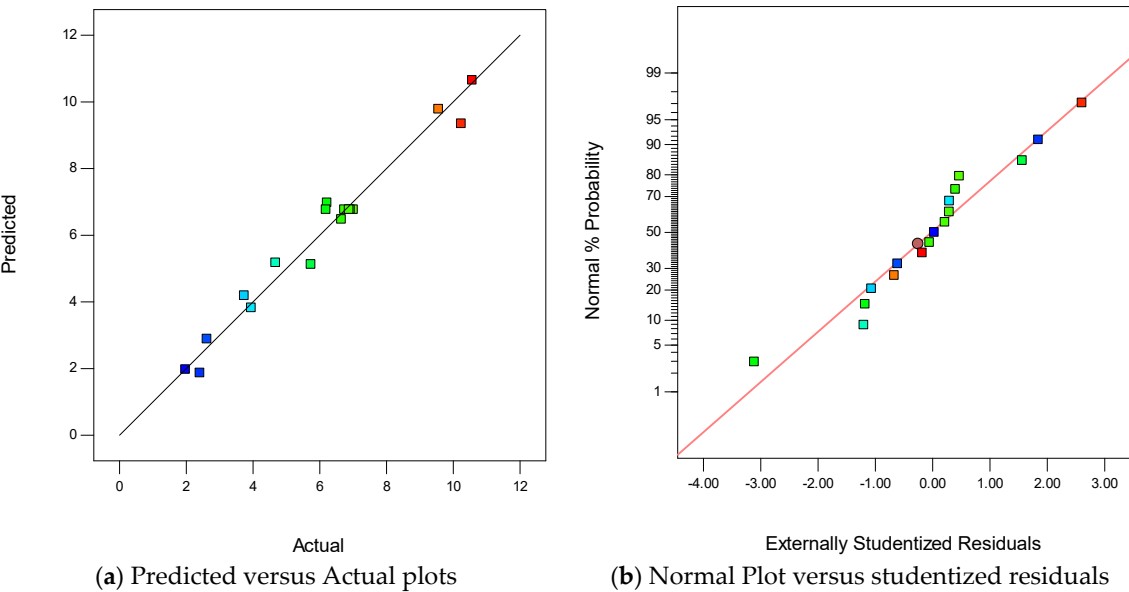

(**a**) Predicted versus Actual plots  (**b**) Normal Plot versus studentized residuals

**Figure 5.** Diagnostic plots for mass loss model.

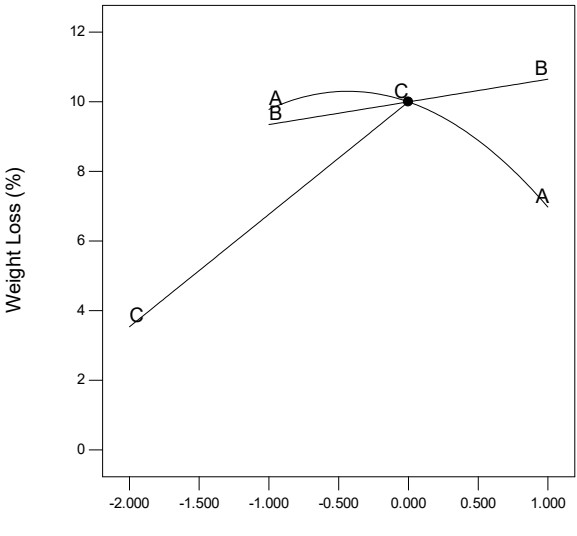

**Figure 6.** Perturbation plot for mass loss.

### 3.1.3. Effect of PAC and Temperature on Mass Loss

The mass loss is used to assess the changes in the physical state of the concrete, mortar, or any other cementitious material due to deterioration after high-temperature exposure [46]. The effect of both DPF, PAC, and temperature on the mass loss of the DPF-reinforced concrete is presented in Figure 7a–c at 300 °C, 600 °C, and 900 °C respectively. The mass loss escalated with an increase in temperature, where the mass loss at 900 °C was the highest (worst) while the mass loss at 300 °C was the lowest (less severe). This is due to the continued deterioration and degradation of both the cement paste, aggregate, and fiber if present in the concrete microstructure. When the concrete microstructure/matrix deteriorated, its losses its structural integrity and easily spalls, this led to an increment in mass loss. At the initial stage, the mass loss resulted from the continuous evaporation of capillary water and gel from the microstructure of the concrete. Further subjection to high temperature led to a loss of interlayer and absorbed water, higher degradation of the concrete matrix, and hence increased mass loss [8]. The mass loss of the concrete further

escalated with an increase in percentage addition of DPF irrespective of the temperature, thus more pronounced at 900 °C. This is due to the degradation of the DPF due to its organic nature and lignocellulose content. When the fiber degrades it creates additional air voids in the cement matrix thereby increasing mass loss. As the DPF itself when added to the concrete even at normal temperature, it increases the pore volume and voids in the concrete matrix. When subjected to high temperatures, these voids created by the DPF contributed to significant evaporation of both absorbed and interlayer water, hence escalating mass loss. The addition of PAC can be seen to also intensify the mass loss of the concrete at 300 °C, 600 °C, and 900 °C, with the effect more pronounced at higher temperatures. This might be ascribed to the expansion of some compounds such as lime (CaO), Nox, carbon and others present takes place during heating. On cooling, these can cause an expansion in volume, which can consequently cause damage to the microstructural integrity of the concrete and hence increase its mass loss.

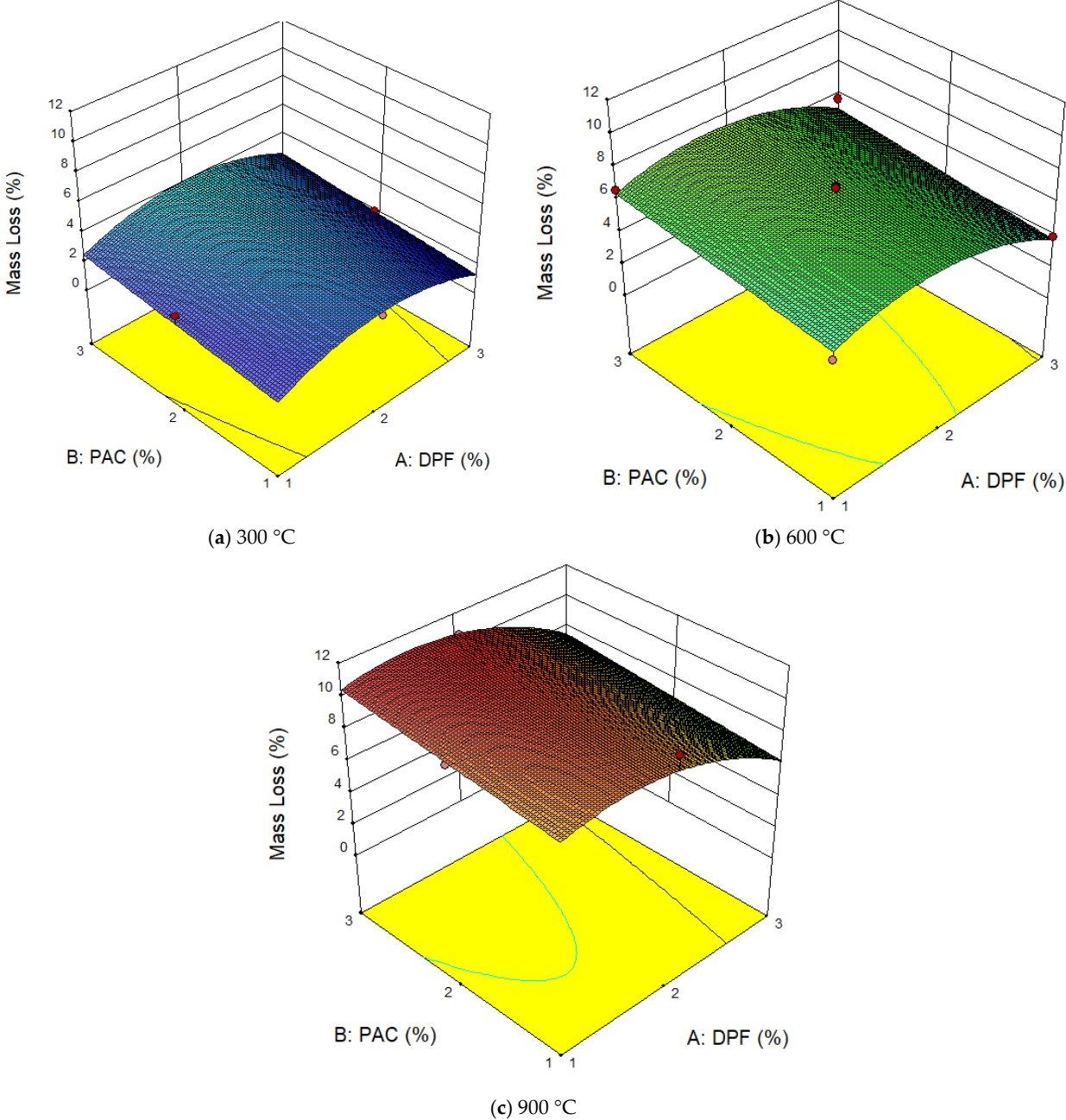

(**a**) 300 °C  (**b**) 600 °C

(**c**) 900 °C

**Figure 7.** 3D response surface plot for mass loss.

### 3.2. Compressive Strength

3.2.1. Analysis of Variance for Compressive Strengths Models

RSM techniques were employed to develop models to predict the residual and relative compressive strengths of the DPF-reinforced concrete when exposed to elevated temperatures through the exploration of the effects of three individual variables (DPF, PAC, and temperature) and their interactive influences. The competence and significance of the model were validated by analysis of variance (ANOVA) of quadratic model type for both compressive and relative strength responses. Table 6 depicts the results of the analysis of variance for the residual and relative strengths models.

**Table 6.** ANOVA summary for compressive strengths models.

| Residual Compressive Strength (MPa) | | | | Relative Strength (%) | | | |
|---|---|---|---|---|---|---|---|
| Source | F Value | *p*-Value Prob > F | Significance | Source | F Value | *p*-Value Prob > F | Significance |
| Model | 28.68 | 0.0001 | Yes | Model | 43.62 | <0.0001 | Yes |
| D-DPF | 19.42 | 0.0031 | Yes | D-DPF | 10.22 | 0.0151 | Yes |
| A-PAC | 15.95 | 0.0052 | Yes | A-PAC | 8.63 | 0.0218 | Yes |
| T-Temperature | 135.15 | <0.0001 | Yes | T-Temperature | 234.00 | <0.0001 | Yes |
| D*A | 10.99 | 0.0128 | Yes | D*A | 10.07 | 0.0156 | Yes |
| D*T | 33.87 | 0.0007 | Yes | D*T | 54.47 | 0.0002 | Yes |
| A*T | 1.36 | 0.2821 | No | A*T | 0.53 | 0.4889 | No |
| $D^2$ | 37.07 | 0.0005 | Yes | $D^2$ | 64.22 | <0.0001 | Yes |
| $A^2$ | 0.79 | 0.4033 | No | $A^2$ | 0.079 | 0.7870 | No |
| $T^2$ | 5.17 | 0.0572 | No | $T^2$ | 12.94 | 0.0088 | Yes |
| Lack of Fit | 0.55 | 0.6731 | No | Lack of Fit | 1.32 | 0.3837 | No |

The F-value (Fisher statistical value) of 28.68 and 43.62 revealed that the models were significant. The value of Prob > F less than 0.05 ($p < 0.05$) also indicated that the models' terms were significant. Moreover, the relative strength model has a lower *p*-value which means that it is more significant compared to the residual compressive strength model. Therefore, the null hypothesis that no relationship existed between the responses (residual compressive and relative strengths) and their variables (DPF, PAC, and temperature) is verified to be false and therefore rejected.

Models developed for predicting the residual compressive strength and the relative strength are given by Equations (5) and (6) where the variables DPF, PAC, and temperature are assigned for simplicity as D, A, and T respectively. The statistical significance of each of the terms is then tested using the confidence interval criteria.

$$F_{C,S} = 86.215 - 38.075^*D + 5.91^*A - 0.053^*T - 3.963^*D^*A + 0.023^*D^*T + 0.0046^*A^*T + 7.092^*D^2 - 1.036^*A^2 - 0.00003^*T^2 \quad (5)$$

$$F_R = 165.46 - 89.54^*D + 18.78^*A - 0.072^*T - 6.78^*D^*A + 0.053^*D^*T + 0.0052^*A^*T + 16.685^*D^2 + 0.585^*A^2 - 0.000083^*T^2 \quad (6)$$

In the equations above, $F_{C,S}$ is the residual compressive strength in MPa, $F_R$ is the relative strength in %. D and A are expressed in terms of percentage and T is given in °C.

For both models, it is noticed that only the terms A*T and $A^2$ have probability values (*p*-values) greater than 0.05. This means that models are not statically significant for these terms. In opposition, all other terms have *p*-values less than 0.05, which indicates strong evidence in contradiction of the null hypothesis. The *p*-value for the lack-of-fit for both models is greater than 0.05, therefore their lack of fits is not significant relative to their pure errors. As a consequence, it is confirmed that the proposed models are well fitted and they adequately describe the practical relationship between the experimental variables and the responses. Additional statistical validation using regression analysis on proposed models as presented in Table 7, showed high $R^2$ values of greater than 0.8 for all responses.

In fact, recorded values of 0.974 and 0.983 for the $R^2$ values for residual compressive and relative strength respectively implied that only 2.6% and 1.7% of the total experimental data were not well fitted into the model. This finding confirms that the generated models were highly correlated with high predictability and were able to clearly explain variability in the responses. Thus, the models are considered as well fitted without any problem and the related equations can be used for prediction without the need for a model reduction or transformation.

**Table 7.** Compressive strengths models' validations.

| Term | Residual Compressive Strength (MPa) | Relative Strength (%) |
|---|---|---|
| $R^2$ | 0.974 | 0.983 |
| Adjusted $R^2$ | 0.94 | 0.960 |
| Predicted $R^2$ | 0.847 | 0.847 |
| Adequate Precision | 22.06 | 26.18 |
| Standard Deviation | 2.39 | 4.27 |
| Mean | 22.81 | 54.26 |
| C.V. % | 10.48 | 7.87 |

Additional ANOVA checks based on standard deviation relative to mean value, coefficient of variation, and adequate precision were performed. For both models, standard deviations of 2.39 and 4.27 compared to means of 22.81 and 54.26 indicates that values are clustered more closely to the mean implying that less variability is present in the experimental data relative to the models. In addition to that, the adequate precision (AP) test performed showed that all the models have AP greater than 4. It is known that AP higher than 4, revealed that the proposed compressive strength models could be used effectively to navigate the design space [45].

### 3.2.2. Diagnostic Plots for Compressive Strengths Models

The predicted versus actual plots are one of the diagnostic plots in RSM used to check the degree of correlation and accuracy of the model graphically. It is also used to check how well the data approximately fit into the fit model. The predicted versus actual plots for the residual compressive strength and relative strength models are illustrated in Figures 8a and 8b respectively. It is perceived that a good correlation, with a very high degree of accuracy, existed between the experimental data and the models as the data points for both residual and relative strength models were very closely aligned across the reference trend line.

The normal against studentized plots for the residual and relative strengths models are shown in Figures 9a and 9b respectively. It is observed that all data points for both models fall close and around the straight normal distribution line. Thus, the assumption on which the design of the experiment and the modeling process is based is verified, which means that the data and models followed a normal distribution.

The perturbation plots for the residual compressive strength and relative strength models are presented in Figure 10a,b respectively. The perturbation plots measure how a variable moves beyond the reference point while other variables are kept constant at that location [45,47]. In Figure 10, A, B, and C denote the variables DPF, PAC, and temperature correspondingly. Line A is more sensitive to both compressive and relative strengths as it shows larger curvature compared to B and C lines. Line B representing the PAC effect is the flattest of the three studied variables and is then characterized as less sensitive to the variation of the compressive strengths of the concrete after exposure to high temperatures.

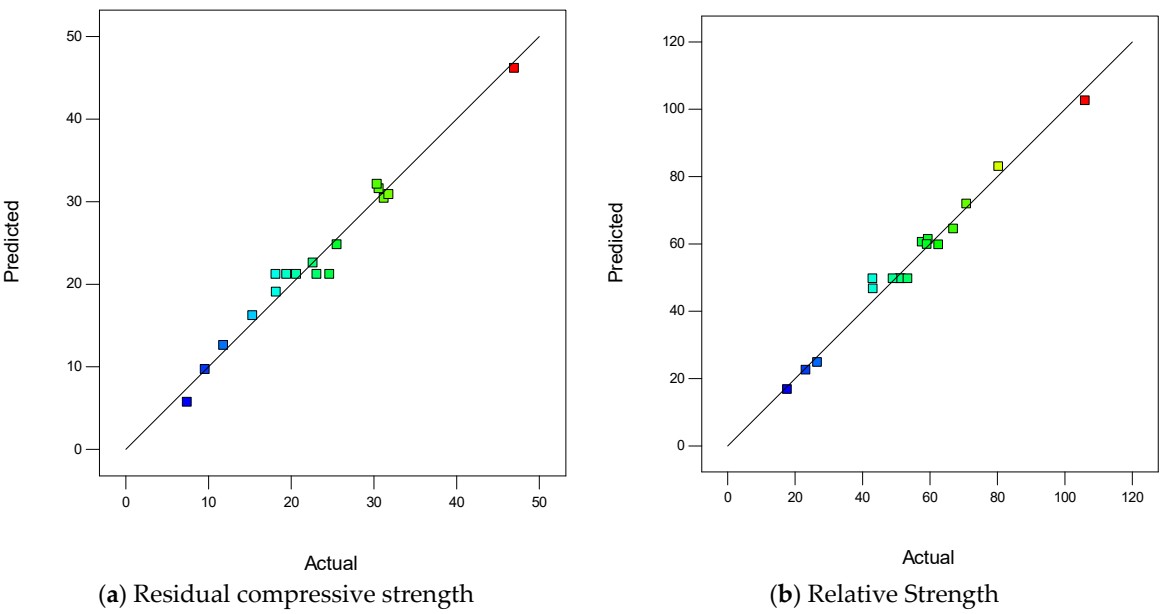

**Figure 8.** Predicted versus Actual plots for compressive strength models.

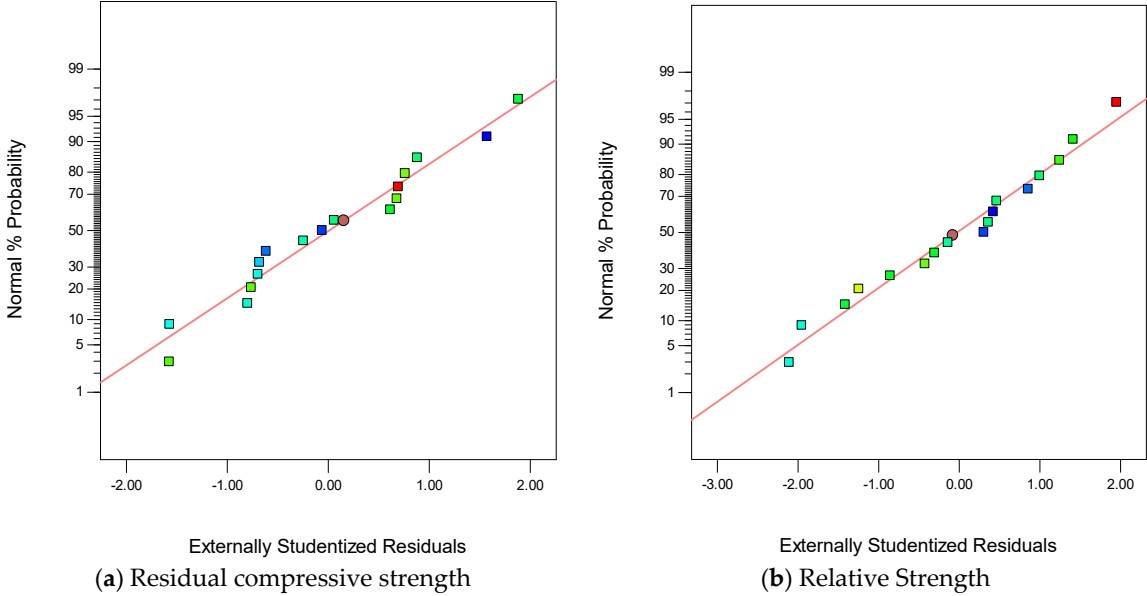

**Figure 9.** Normal plots against studentized residuals for compressive strength models.

### 3.2.3. Effect of PAC and Temperature on Compressive Strengths

As defined earlier in this study, the residual compressive strength is the maximum load per unit area the concrete can withstand after heat exposure, while the relative strength is the percentage variation between the compressive strength of the concrete at normal temperature and its compressive strength after heating. Figure 11a–c presents the 3D response surface plots for residual compressive strengths at 300 °C, 600 °C, and 900 °C respectively. While Figure 12a–c shows the response surface plot for the relative strengths of the concrete for the same range of selected temperatures. The compressive strength declined with an increase in temperature as shown in Figure 11. Such a reduction might be explained by the fact that when the higher exposure to temperature, the formation of micro-cracks and the degradation of the cement matrix increases. Thus, this will lead to a high loss in compressive strength. At 300 °C, the addition of PAC resulted in a noticeable increase of both residual compressive strength and relative strengths as displayed in

Figures 11a and 12a. Similarly, at 600 °C, a slight increase in compressive strength was noticed which was less pronounced compared to 300 °C exposure. This enhancement in strength can be ascribed to the formation of excess tobermite in the cement matrix as a result of the hydrothermal reaction between the calcium oxide and unhydrated silica in the cement and other chemical elements in the PAC constituent [8]. Moreover, this could be explained also by the increase in the Van der Waals forces intensity between the gel particles resulting from the evaporation of moisture under elevated temperature [48,49]. The addition of DPF can be seen to cause a reduction in both residual compressive strength and relative strength, where the reduction is more pronounced at higher temperatures. The cause could be attributed to the increase in pore volume by the DPF which speed up the widening of existing micro-cracks and the creation of new micro-cracks under the effect of heat, this consequently leads to a decline in strength. The DPF under high temperature conjointly degrades and causes thermal expansion in the cement matrix resulting in weakening the structural integrity and consequently in strength.

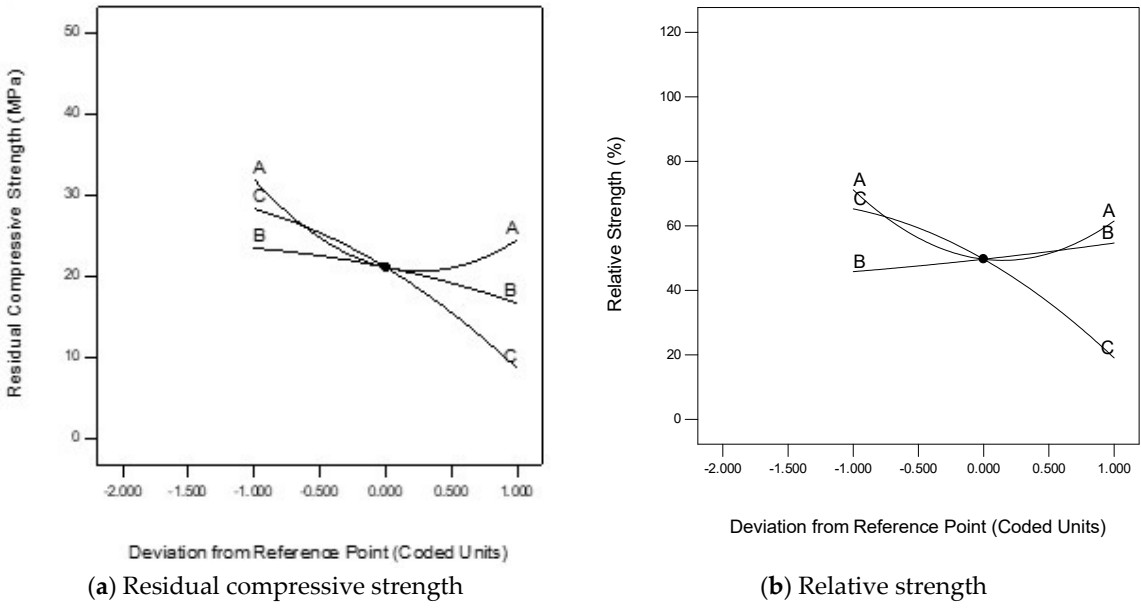

(**a**) Residual compressive strength　　　　　(**b**) Relative strength

**Figure 10.** Perturbation plots for compressive strengths.

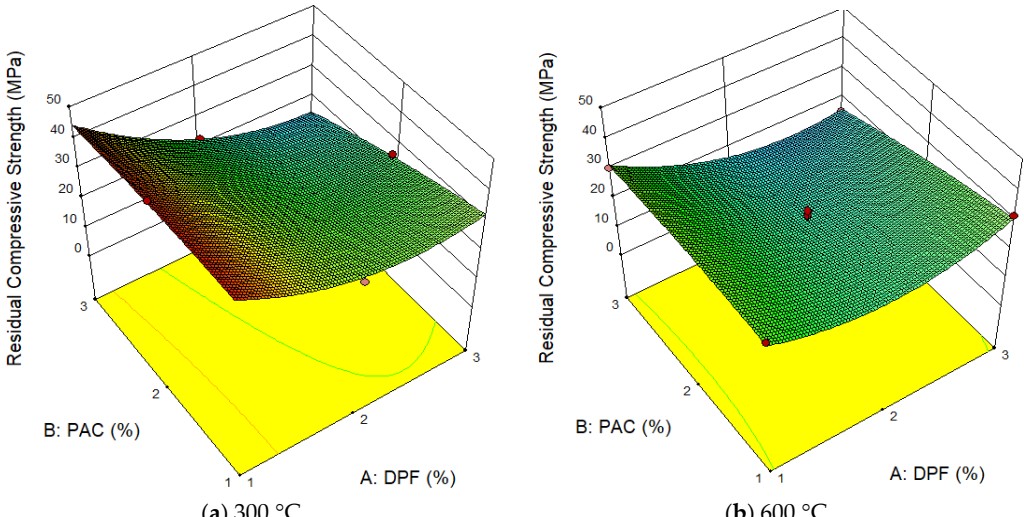

(**a**) 300 °C　　　　　(**b**) 600 °C

**Figure 11.** *Cont.*

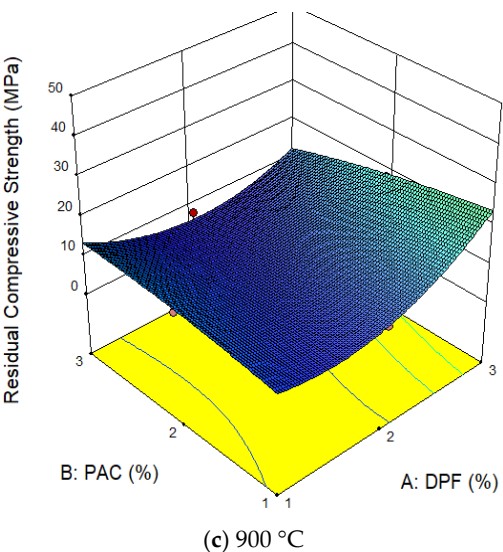

(**c**) 900 °C

**Figure 11.** 3D Response surface plot for Residual compressive strength.

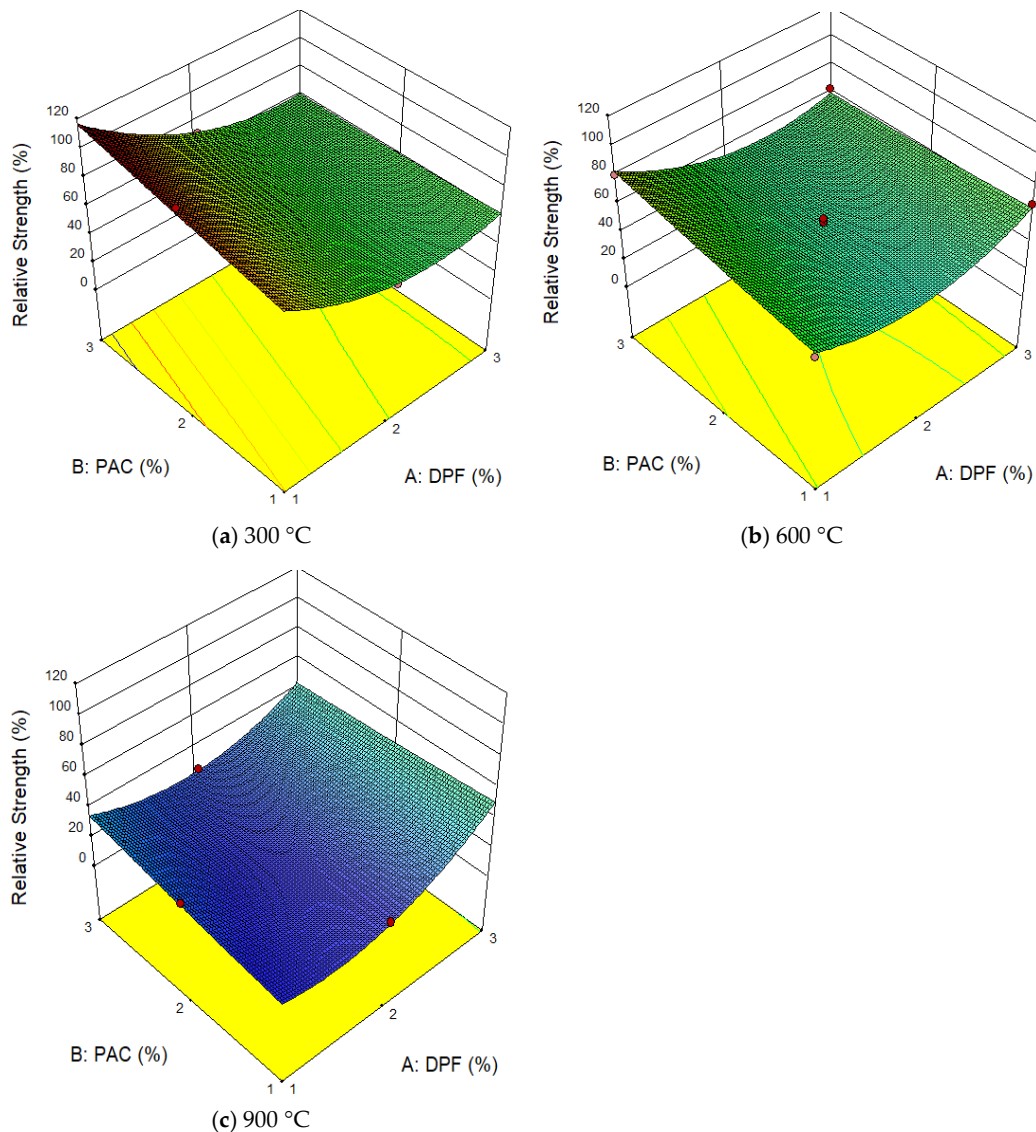

**Figure 12.** 3D Response surface plot for Relative compressive strength.

### 3.3. Multi-Objective Optimization

A multi-objective optimization using the RSM concept was carried out to accomplish the optimal performance of DPF-reinforced concrete modified with PAC under elevated temperature exposure, in terms of lower mass loss and higher residual compressive strength and relative strength.

In the optimization criteria, the variables DPF, PAC, and temperature are kept within the range, while the mass loss was minimized, and strengths were maximized.

The optimization results are presented in Table 8. As observed, the DPF-reinforced concrete produced using 1% DPF, 2.27% PAC and subjected to an elevated temperature of 300 °C; yielded the lowest mass loss of 2.05%, the highest residual compressive strength and relative strength of 45.85 MPa and 106.7% respectively with the desirability of 98.8%. For a perfect and accurate optimization, its desirability will be 100%, hence the results of the optimization in this study can be said to have excellent desirability and accuracy. Therefore, to achieve the best performance of the DPF-reinforced concrete under high temperatures, it is recommended to produce the concrete by adding 1% DPF, 2.27% PAC by weight of cement and other conventional constituents (Table 2) and heat the concrete to a temperature of 300 °C for a period of two hours.

**Table 8.** Optimization Conditions and Solutions.

| Variables/Responses | Goal | Lower Limit | Upper Limit | Solutions | | |
|---|---|---|---|---|---|---|
| | | | | No. 1 | No. 2 | No. 3 |
| D: DPF (%) | In range | 1 | 3 | 1.0 | 1.0 | 1.0 |
| A:PAC (%) | In range | 1 | 3 | 2.27 | 2.31 | 2.32 |
| T: Temperature (°C) | In range | 300 | 900 | 300 | 305.4 | 311.3 |
| Mass loss (%) | Minimize | 1.98 | 10.57 | 2.05 | 2.14 | 2.23 |
| Residual Compressive Strength (MPa) | Maximize | 7.42 | 47.01 | 45.85 | 45.60 | 45.36 |
| Relative Strength (%) | maximize | 17.73 | 106.07 | 106.7 | 106.1 | 105.8 |
| Desirability (%) | | | | 98.8 | 98.2 | 97.5 |

### 3.4. Model Validation

The developed models for predicting the mass loss and strengths of the DPF-reinforced concrete were validated practically by conducting a series of additional experiments in the laboratory. The optimal solutions in Table 8 and other variable combinations were used together with the constant constituent materials in Table 2 to produce the mixes and subject them to the appropriate temperatures before testing. The same variable proportions were used in the developed models to estimate the mass loss and strengths of the DPF-reinforced concrete. The percentage error between the estimated and experimental responses was calculated using the following Equation (7).

$$\xi = \frac{E - P}{E} \times 100 \tag{7}$$

where $\xi$ denotes the error in %, *E* represents the experimental results, and *P* is the estimated response using the models.

The results of the additional model's experimental validation are summarized in Table 9. All the models have an acceptable error of less than 8%. The mass loss has a mean error of 6.79%, while the residual and relative compressive strengths have average errors of 6.66% and 7.06% respectively, which is very satisfactory as a result. Therefore, the established models for estimating the mass loss and compressive strengths of DPF-reinforced concrete containing DPF and PAC under high temperatures are considered highly accurate and well-predictive.

**Table 9.** Models' Experimental Justification.

| Responses | Variables | | | Predicted | Experimental | Error | Mean Error (%) |
|---|---|---|---|---|---|---|---|
| | DPF | PAC | Temp | | | | |
| Mass loss | 1 | 2.27 | 300 | 2.03 | 2.16 | 5.84 | |
| | 1 | 2.32 | 311.3 | 2.22 | 2.1 | 5.48 | |
| | 2 | 1 | 600 | 6.10 | 6.52 | 6.52 | |
| | 3 | 1 | 900 | 6.29 | 6.93 | 9.31 | 6.79 |
| | 1 | 1 | 400 | 2.52 | 2.75 | 8.22 | |
| | 1.5 | 2.5 | 700 | 8.20 | 8.67 | 5.38 | |
| Residual Compressive Strength | 1 | 2.27 | 300 | 45.75 | 43.27 | 5.72 | |
| | 1 | 2.32 | 311.3 | 45.25 | 47.19 | 4.11 | |
| | 2 | 1 | 600 | 23.14 | 25.3 | 8.53 | |
| | 3 | 1 | 900 | 23.04 | 21.46 | 7.38 | 6.66 |
| | 1 | 1 | 400 | 41.18 | 43.5 | 5.33 | |
| | 1.5 | 2.5 | 700 | 18.90 | 20.74 | 8.88 | |
| Relative Strength | 1 | 2.27 | 300 | 113.23 | 105.6 | 7.23 | |
| | 1 | 2.32 | 311.3 | 113.39 | 120.2 | 5.66 | |
| | 2 | 1 | 600 | 52.57 | 57.3 | 8.26 | |
| | 3 | 1 | 900 | 61.78 | 57.3 | 7.82 | 7.06 |
| | 1 | 1 | 400 | 86.39 | 91.6 | 5.69 | |
| | 1.5 | 2.5 | 700 | 67.55 | 73.2 | 7.72 | |

## 4. Conclusions

In this research, RSM has been employed to design experiments, establish models, and investigate the effect of powdered activated carbon (PAC) addition on mass loss and compressive strengths of the DPF-reinforced concrete subject to elevated temperature exposure. The following conclusions could be deduced from the present work:

(1) The addition of DPF and PAC escalated the mass loss of the DPF-reinforced concrete under high temperatures.

(2) DPF addition led to a significant loss in compressive strength of the concrete under high temperatures, where the loss in strength was more pronounced when the concrete was subjected to a higher temperature above 300 °C.

(3) PAC incorporation as additive by weight of cement to the DPF reinforced concrete mitigated the loss in strengths of the concrete when subjected to high temperatures up to 600 °C. This was noticed as the addition of PAC improved the residual and relative strengths of the DPF-reinforced concrete when heated to a temperature of up to 600 °C.

(4) The mathematical models developed using RSM to predict the performance of the DPF-reinforced concrete under high temperatures in form of weight loss, relative strength, and residual compressive strengths were all very significant. The models have a very high degree of correlation and predictability. These models were validated experimentally and found to have an average error of less than 8%.

(5) The highest compressive and relative strengths and lowest mass loss for the DPF under high temperature were achieved using the optimum combinations of 1% DPF and 2.27% PAC as additive by weight of cementitious materials, subjected to a temperature of 300 °C for 2 h. exposure.

(6) It is recommended to use PAC as an additive in natural fiber-reinforced concrete under high temperatures to partially mitigate the loss in strength of the concrete.

**Author Contributions:** Conceptualization, M.A., Y.E.I. and H.A.; methodology, M.A. and H.A.; software, M.A., O.E. and H.A.; validation, M.A. and N.M.A.; formal analysis, M.A., O.E. and H.A.; investigation, M.A., Y.E.I. and H.A.; resources, Y.E.I. and H.A.; data curation, O.E. and N.M.A.; writing—original draft preparation, M.A. and H.A.; writing—review and editing, Y.E.I., O.E. and

N.M.A.; visualization, N.M.A.; supervision, Y.E.I.; funding acquisition, Y.E.I. and H.A.; project administration: H.A. All authors have read and agreed to the published version of the manuscript.

**Funding:** The authors extend their appreciation to the deputyship for Research and Innovation, Ministry of Education in Saudi Arabia for funding this research work through the project number (IFP-2022-28).

**Institutional Review Board Statement:** Not applicable.

**Informed Consent Statement:** Not applicable.

**Data Availability Statement:** Data will be available on request.

**Acknowledgments:** The authors extend their appreciation to the deputyship for Research and Innovation, Ministry of Education in Saudi Arabia for funding this research work through the project number (IFP-2022-28).

**Conflicts of Interest:** The authors declare no conflict of interest.

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
