# Peer review of "Modeling and Optimization of Date Palm Fiber Reinforced Concrete Modified with Powdered Activated Carbon under Elevated Temperature"

_sustainability, doi:10.3390/su15086369_

Round 1
Reviewer 1 Report
The paper presents data on the use of DPF and PAC to enhance the properties of concrete. The attached paper has suggestions to improve the quality of English - these are generally minor and quick edits.
The paper would benefit from the addition of references to the literature on wood fibre addition to concrete. There are reviews on this subject, and DPF is a subset of wood fibres and lignocellulosic materials.
The paper does not suggest an optimum composition. It would benefit from an indication of the range of preferred compositions, in the conclusions.

Author Response
Thank you for taking your precious time to review our manuscript. All your comments were carefully considered as can be seen in the attached file.

Reviewer 2 Report
The paper shows the effect of the correlation between the date palm fiber and powdered activated carbon on the relative compressive strength in concrete during exposure to different elevated temperatures using toll "response surface methodology. It is well structured and written. however, there are minor mistakes and need to be corrected:
1) Table 2 and 3: it used unity (%) for temperature instead of ° under Variables.
2) Lines 296, 302: corelated wrong spelling
Author Response

(The authors gave the same response as above.)

Reviewer 3 Report
Dear Editor: For the study to be accepted, the author (s) must respond to the following comments point by point to reach the level of a high-quality journal.
1. The followings are my comments regarding this manuscript. The authors should briefly discuss their innovation in the abstract, which has to be improved with much more information.
2. The abstract needs to be improved by highlighting this paper's key findings.The entire abstract section must be revised to briefly explain this research study's importance, investigations, and outcomes with advantages/significance.
Abstract :
3. The entire abstract section must be revised to briefly explain this research study's importance, investigations, and outcomes with advantages/significance.
4. Nothing is mentioned in the abstract regarding the mixed proportion, experimental study, and the range of parameters used. Please revise it.
5. Resent a detailed graphical abstract for this work, which could be more interesting for the reader community. The novelty of the study should be reflected in the abstract.
Introduction:
6. The introduction section is not up to the mark. You only need to connect state-of-the-art to your paper goals in the introduction section. Hence modify the entire section accordingly and present the specific goals/research objectives in the last part of the introduction section.
7. The research significance has to be highlighted at the end of the introductions.
8. The authors must add more information and supported studies to the introduction since the introduction is poor and needs to be strengthened. Recent publications in the area of natural fiber in concrete, elevated temperature effects, compressive strength modeling for concrete, and sensitivity analysis for the developed models have to be explained in the introduction to justify the study; the following references have to be considered in the study :
• Evaluation of Mechanical and Permeability Characteristics of Microfiber-Reinforced Recycled Aggregate Concrete with Different Potential Waste Mineral Admixtures
• Improving the performance of recycled aggregate concrete using nylon waste fibers
Methodology:
1. A flowchart should be provided for the work process. The flow chart of the study has to be described in the steps.
2. Many grammatical errors need to be corrected. Several grammar errors can be observed in the paper, which is negatively affected by the paper's quality.
3. The frequency of the device has to be mentioned in the study. The properties of the devices used in the study, such as strain rate, machine deflection, and capacity, have to be mentioned in the methodology,
4. The gravel and sand grain size distribution must be compared with upper and lower, followed by ASTM standards (Fig.1), and the y-axis for fig. 1 must be stopped at 100%. Please fix it.
Results and discussion:
5. RMSE and MAE have to be also calculated, and it can not evaluate the model performance only using R2.
6. Major comment: The study is missing the sensitivity analysis to evaluate the effect of each model parameters such as temeperature, PAC, DPF, cement , aggregated and w/c, similar to this paper " Surrogate Models to Predict the Long-Term Compressive Strength of Cement-Based Mortar Modified with Fly Ash" The most affected variables on the compressive strength of concrete have to be found based on the model outcomes?
7. Without witness lab photos, the results are not believable. The shape of the sample failures and the paths have to be provided in the study.
8. The model outcomes are missing the error percentage lines. No Error % was found for the models.
9. Major comment: The authors must evaluate the model performance in different strength ranges. It is very well known that the composition of the materials is different in, for example, in compressive strength range of 10- 50 MPa than the materials with compressive strength of 50 MPa. Use this study to evaluate the model performance in the different strength ranges “Analysis and prediction of the effect of Nanosilica on the compressive strength of concrete with different mix proportions and specimen sizes using various numerical approaches.” Similar strength range model evaluations must be done and support the procedure using the mentioned study.
10. The compressive strength of concrete is not only dependent on time; it is multivariable functional such as w/c, sand, gravel, cement contents, temperature, fiber content, and the shape-size of the samples. The authors have to elaborate and compare the results with the results in this article.
• The effectiveness of surrogate models in predicting the long-term behavio

Author Response

(The authors gave the same response as above.)

Round 2
Reviewer 3 Report
Dear Editor:
The authors carefully studied the reviewer's comments and revised the manuscript. In my opinion, this manuscript's quality meets the journal's requirements. I suggest this manuscript be accepted and published in this journal.